# A Polysaccharide Purified from *Morchella conica*
*Pers.* Prevents Oxidative Stress Induced by H_2_O_2_ in Human Embryonic Kidney (HEK) 293T Cells

**DOI:** 10.3390/ijms19124027

**Published:** 2018-12-13

**Authors:** Na Xu, Yi Lu, Jumin Hou, Chao Liu, Yonghai Sun

**Affiliations:** 1College of Food Science and Engineering, Jilin University, Changchun 130062, China; doctorserena@163.com; 2Jilin Provincial Key Laboratory of Animal Embryo Engineering, Jilin University, Changchun 130062, China; luyi920127@163.com; 3College of Food Science and Engineering, Changchun University, Changchun 130028, China; houjumin0511@126.com; 4School of Food Engineering, Jilin Agriculture Science and Technology College, Jilin 132101, China; liuchaocarrol@163.com

**Keywords:** *Morchella conica*, polysaccharides, oxidative stress

## Abstract

*Morchella conica Pers.* (*M. conica*) has been used both as a medical and edible mushroom and possesses antimicrobial properties and antioxidant activities. However, the antioxidant properties of polysaccharides purified from *M. conica* have not been studied. The aim of this study was to investigate the in vitro antioxidant properties of a polysaccharide NMCP-2 (neutral *M. conica* polysaccharides-2) purified from *M. conica,* as determined by radical scavenging assay and H_2_O_2_-induced oxidative stress in HEK 293T cells. Results showed that NMCP-2 with an average molecular weight of 48.3 kDa possessed a much stronger chelating ability on ferrous ions and a higher ability to scavenge radical scavenging 2,2-diphenyl-1-picrylhydrazyl (DPPH) than the other purified fraction of NMCP-1 from *M. conica*. Moreover, 3-(4, 5-Dimethylthiazol-2-yl)-2, 5-diphenyltetra-zolium bromide (MTT) assay showed that NMCP-2 dose-dependently preserved cell viability of H_2_O_2_-induced cells. The NMCP-2 pretreated group reduced the generation of reactive oxygen species (ROS) content and increased the mitochondria membrane potential (MMP) levels. In addition, Hoechst 33342 staining revealed cells treated with NMCP-2 declined nuclear condensation. Ultrastructural observation revealed that NMCP-2 pretreatment alleviated the ruptured mitochondria when exposed to H_2_O_2_. Furthermore, western blot analysis showed that NMCP-2 prevented significant downregulation of the protein expression of Bax, cleaved caspases 3, and upregulated Bcl-2 levels. These results suggest the protective effects of NMCP-2 against H_2_O_2_-induced injury in HEK 293T cells. NMCP-2 could be used as a natural antioxidant of functional foods and natural drugs.

## 1. Introduction

Oxidative stress has been implicated in several chronic diseases that include aging, cancer, diabetes, cardiovascular diseases, and neurodegenerative diseases [1]. Oxidative stress is a condition of imbalance between prooxidants and antioxidants, which is mainly caused by the excessive accumulation of reactive oxygen species (ROS), such as hydrogen peroxide (H_2_O_2_), hydroxyl free radicals (^•^OH), and hydroxyl free radicals (O_2_^−^) [2]. The excessive ROS can pass through the cell membrane and cause oxidative damage to lipids, proteins, and DNA, thereby inducing cell apoptosis through different metabolic pathways [1,3]. Thus, the scavenging of ROS mediated by antioxidants might be a promising treatment option for retarding oxidative stress-related diseases [4].

Edible fungi or mushrooms are rich in nutrition, and possess notable medicinal properties and bioactivities [5]. The most abundant biological macromolecules in edible fungi are polysaccharides. They have a wide range of biological activities, including antitumor [6], antimicrobial [7], immunomodulatory [8], prebiotic [9], and antioxidant activities [10]. *Morchella conica Pers.* (*M. conica*) is a rare edible mushroom belonging to the genus *Morchella,* well known for its delicate taste and unique appearance [4,11]. Previous studies have showed the *M. conica* extracts play an important role in scavenging free radicals [12]. However, reports about the antioxidant activities of purified polysaccharides from *M. conica* and the evaluation of their effects on preventing oxidative stress are barely mentioned.

In the present study, we analyzed the chemical composition and preliminary structural features of purified MCP (*M. conica* polysaccharides) fraction of neutral *M. conica* polysaccharides-2 (NMCP-2). We investigated the protective effect of NMCP-2 on H_2_O_2_-induced oxidative stress in HEK 293T cells and analyzed its effects on cell viability, the generation of ROS, apoptosis, and the mechanisms in vitro. 

## 2. Results and Discussions

### 2.1. Purification of Crude MCP

The crude MCP was separated through a DEAE-52 cellulose column, fractionated into two polysaccharide peaks designated as NMCP, AMCP (acidic *M. conica* polysaccharides) (Figure 1a). The main fraction (NMCP) was collected and further purified with Sephadex G-100 gel filtration chromatography, affording two independent elution peaks of NMCP-1 and NMCP-2 (Figure 1b). In this study, NMCP-1 and NMCP-2 were collected for further radical scavenging analysis.

### 2.2. DPPH (2,2-diphenyl-1-picrylhydrazyl) Scavenging Effect and Ferrous Ion Chelating Ability of NMCP-1 and NMCP-2

DPPH is a stable free radical that has been extensively used for free radical elimination reactions. Free radicals are scavenged when they encounter an electron or hydrogen donor [13]. It can be seen from Figure 2a that the DPPH radical scavenging abilities of NMCP-1 and NMCP-2 were dose-dependent when comparison with the same concentrations of Vitamin c (Vc). At the concentration of 4 mg/mL, the scavenging activities of NMCP-1 and NMCP-2 are 48.29 ± 4.61% and 73.49 ± 6.14%, respectively. The DPPH scavenging ability in NMCP-2 at six concentrations from 0.1 to 4 mg/mL was significantly stronger than that in NMCP-1 groups at the same concentrations (*p* < 0.05). Compared with other polysaccharides purified from fungi, the DPPH scavenging ability of NMCP-2 is similar to GFP-2 (*Grifola Frondosa* polysacchaeide-2) purified from *Grifola frondosa*. At the concentration of 3 mg/mL, the DPPH scavenging ability of GFP-2 and NMCP-2 were 79.6% and 70%, respectively [14]. The DPPH scavenging ability of NMCP-2 is higher than that of the polysaccharides purified from *Penicillium* sp. F23-2 [15]. In our study, it was found that NMCP-1 and NMCP-2 were hydrogen donors to the DPPH free radicals, thereby terminating the radical chain reaction. NMCP-2 showed a remarkably better scavenging capacity than NMCP-1 at the dosage of 0–4 mg/mL.

Ferrous is the strongest prooxidant that stimulates the lipid peroxidation among transition metals. Hence, the Fe^2+^ chelating capacity was applied to antioxidant research via a measurement of the iron-ferrozine complexes [16]. As shown in Figure 2b, the Fe^2+^ chelating rates of purified NMCP-1 increased from 15.02% to 90.15% when the concentration increased from 0.1 to 4.0 mg/mL. For NMCP-2, the chelating ferrous ability increased from 18.24% to 93.08% as the concentration increased from 0.1 to 1.0 mg/mL, and slightly increased when the concentration of NMCP-2 was increased from 2.0 to 4.0 mg/mL. The Fe^2+^ chelating capacity in NMCP-2 in the 0.5 and 1 mg/mL groups was significantly stronger than that in NMCP-1 groups at the same concentrations (*p* < 0.05). Furthermore, the NMCP-2 possessed superior binding capacity for Fe^2+^ than NMCP-1. NMCP-2 was also more effective in its chelating ability than other fungi polysaccharides, such as from *Tricholoma matsutake* [17]. Because NMCP-2 possessed both a higher antioxidant activity of DPPH scavenging and better ferrous ion chelating ability than NMCP-1, NMCP-2 was selected for the subsequent assay [18,19].

### 2.3. Chemical Characters of the Polysaccharide 

The characteristic organic groups in the polysaccharide were identified by FT-IR. Bands around 3400, 2920, 1620, 1400, and 1100 cm^−1^ are the characteristic absorption peaks of polysaccharides [20]. The FT-IR spectrum of NMCP-2 is shown in Figure 3a. The strong extensive absorption at 3428 cm^−1^ was from the hydroxyl stretching vibration. A weak peak at 2925 cm^−1^ was due to C-H asymmetric stretching vibration. The bands at 1627 cm^−1^ and 1384 cm^−1^ indicated the stretching vibration of the carboxyl group [21]. The peak at 1081 cm^−1^ was associated with pyranose rings.

The average molecular weight of NMCP-2 was measured by HPLC (High-performance liquid chromatography). As is shown in Figure 3b, NMCP-2 appeared as a single symmetrical elution peak, indicating a homogeneous polysaccharide. The molecular weight was estimated by a calibration curve generated by standard dextran T-series. The average molecular weight of NMCP-2 was estimated to be 48.3 kDa.

The gas chromatography-mass spectrometry (GC–MS) method was used to identify the monosaccharide compositions of the polysaccharide. The standard monosaccharides are mannose (Man), rhamnose (Rha), glucose (Glu), arabinose (Ara), xylose (Xyl) and galactose (Gal). The GC-MS spectrum is shown in Figure 3c. The composition of the polysaccharide was identified by referencing the retention times with those of the standard monosaccharides. NMCP-2 was composed of mannose, glucose, galactose, and xylose in the molar ratio of 3.35:19.57:1.00:3.14.

To further elucidate the structural characteristics of NMCP-2, NMR analysis was implemented (Figure 4) and the ^1^H and ^13^C NMR shifts of NMCP-2 are listed in Table 1. In the ^1^H NMR spectrum, NMCP-2 contained six anomeric signals (5.12, 5.08, 5.02, 4.98, 4.67, and 4.54 ppm). The ^13^C NMR spectrum showed two dominant (99.3 and 98.22 ppm) and five minor (103.43, 102.22, 101.3, 101.15, and 96.24 ppm) anomeric signals. From the HSQC spectrum, seven peaks in the anomeric carbon hydrogen region revealed seven residues in NMCP-2. 

In ^1^H and ^13^C NMR, the anomeric carbon signal at 99.3 ppm and anomeric hydrogen signal at 5.12 ppm indicated that residue A was α-linked. Besides, the downfield shifts of C-4 and C-1 from the HSQC spectrum suggested that residue A was →4)-α-D-Glup-(1→. Therefore, with the same method, the other residues were →6)-α-D-Glup-(1→ (residue B), →4, 6)-α-D-Glup-(1→ (residue C), →3)-β-Man-(1→ (residue E), →4)-β-D-Xylp-(1→ (residue F), and →4)-β-D-Galp-(1→ (residue G), respectively. However, residue F showed no ^13^C shifts caused by glycosylation, which indicated residue F was α-T-Man.

The sequences between glycosyl residues were determined by HMBC spectrum. The cross-signal of G H1- F C4 indicated that the residue G was linked to the O-4 position of residue F. The cross peak of F H1-B C6 showed that residue F was attached to the O-6 position of residue B. The signal of B H1- A C4 suggested that residue B was attached to the O-4 position of residue A, which was ascertained by the signal of A H4- B C1. Furthermore, the cross peaks of B H1- C C6, C H4- A C1, C H1- E C3, and A H4- D C1 were observed. Combining the above results, the possible fragments of the NMCP-2 sequence are shown in Figure 5.

### 2.4. Effects of NMCP-2 on the Viability of HEK 293T Cells

Maintaining redox homeostasis is essential for cellular physiology. Oxidative stress disrupts redox homeostasis and shifts to an oxidized condition caused by the excessive ROS, resulting in the cell damage. Among the ROS, H_2_O_2_ diffuses through cell membranes and generates hydroxyl radicals through the Fenton reaction [22,23], inducing redox changes and apoptosis [24]. Therefore, H_2_O_2_ is appropriate to establish an oxidative damage model for studying the protective effects of antioxidants.

The cytotoxicity of NMCP-2 on HEK 293T cells was evaluated by 3-(4, 5-Dimethylthiazol-2-yl)-2, 5-diphenyltetra-zolium bromide (MTT) (Figure 6a). Incubation with 0–50 µg/ml NMCP-2 did not exhibit any significant effects on the viability and proliferation of the cells. However, further treatment of HEK 293T cells with 400 µM H_2_O_2_ resulted in a response in a dose-dependent manner compared with the control group (Figure 6b). The cell viability increased from 46.34% to 93.77% when the concentration of NMCP-2 increased from 5 to 25 µg/mL. While at the concentration of 50 µg/mL, the cell viability was 90.15%, which was close to that in 25 µg/ml treatment. The results indicated that NMCP-2 could protect the cells from oxidative stress. Thus, further analysis employed solutions with concentrations of 5, 10, and 25 µg/mL NMCP-2.

### 2.5. Protective Effect of NMCP-2 against H_2_O_2_-Induced HEK 293T Cells Apoptosis

To determine whether NMCP-2 shows a potent ROS-scavenging effect in HEK 293T cells, the levels of ROS were evaluated using the fluorescence probe DCFH-DA (Dichloro-dihydro-fluorescein diacetate). As is shown in Figure 7a, the cells exposed to H_2_O_2_ displayed an increased level of fluorescence intensity by comparison with that in the control group. Pretreatment with NMCP-2 significantly inhibited the up-regulation of fluorescence intensity in H_2_O_2_-induced HEK-293 cells. 

Mitochondria play an important role in regulating the apoptosis pathway. Mitochondria are both important sources of ROS-generation and a major target of ROS invasion [25]. Excessive accumulation of ROS may activate mitochondrial stress pathways and cause mitochondrial injury. We detected the mitochondria membrane potential intensity on HEK 293T cells after the treatment of NMCP-2 and H_2_O_2_ (Figure 7b). Results shows that exposure of HEK 293T cells to H_2_O_2_ induced a significant reduction in the mitochondria membrane potential (MMP) (44.69%) compared with that (100%) in the control group. Pretreated cells with NMCP-2 (5, 10, and 25 μg/mL) significantly enhanced the MMP, suggesting a role for NMCP-2 in the protection of the mitochondria during oxidative stress. NMCP-2 at the concentration of 25 μg/mL showed a significant effect on preventing cell apoptosis. Thus, 25 μg/mL of NMCP-2 was selected for subsequent analysis.

To further determine the inhibitory effects of NMCP-2 on cell apoptosis, Hoechst 33342 staining was performed to examine the morphological changes in HEK 293T cells by fluorescence microscopy. The normal cells were stained with faint blue fluorescence. While apoptotic cells were stained with bright blue fluorescence. As shown in Figure 8, the nuclei in the control group were round, regular, and homogeneously stained with faint blue fluorescence (Figure 8a). The apoptosis cells treated by H_2_O_2_ appeared to have cell shrinkage and nuclear condensation, displaying bright blue fluorescence (Figure 8b). The number of apoptotic cells with irregular morphology and chromatin condensation was decreased via pretreatment with NMCP-2 (Figure 8c,d).

### 2.6. Effects of the NMCP-2 on Ultrastructure Changes Caused by H_2_O_2_

TEM was applied to observe the ultrastructure of HEK 293T cells. As is shown in the Figure 9, the cells in the control groups exhibited clearly visible nuclear membranes and well-preserved euchromatin of the nucleus (Figure 9a). In addition, the H_2_O_2_-treated cells had chromatin condensation (Figure 9b), ruptured or absent cristae in the mitochondria, and larger lysosomes (Figure 9f). By contrast, incubation with 5 μg/mL NMCP-2 remarkably reduced the number of vacuoles and amount of chromatin condensation compared with that of the H_2_O_2_-treated cells (Figure 9c,g). Treatments with 25 μg/mL NMCP-2 significantly alleviated the pathological changes in mitochondrial swelling in comparison to the H_2_O_2_ group (Figure 9d,h). Compared with the model group, NMCP-2 meaningfully reversed the above conditions.

### 2.7. mRNA Expression of Apoptosis Related Genes after NMCP-2 Exposure

Impaired redox status leads to accumulation of ROS, which activates mitochondrial stress. Apoptosis is triggered when homeostasis of the redox status is disrupted and the adaptive defense system fails to compensate for the stress [26]. Apoptosis is usually activated through two primary pathways: the receptor-mediated signaling pathways (extrinsic pathways) and mitochondrial mediated pathways (intrinsic pathways) [27]. The extrinsic signaling pathway is triggered by cytokine ligands, such as the tumor necrosis factor (TNF) receptor and Fas ligand (FasL), leading to the activation of downstream apoptosis cascades [28]. On the other hand, mitochondrial-mediated apoptosis signaling pathways induce the generation of ROS and the reduction of MMP, regulated by the anti-apoptotic family members (Bcl-2, Bcl-w, Bcl-k, etc.) and pro-apoptotic members (Bax, Bak, Bid, Bim, Bad, etc.) [29]. The anti-apoptotic member Bcl-2 is able to inhibit cell death [30]. The pro-apoptotic effector proteins Bax are regarded as crucial effectors to accelerate the permeability of mitochondria [31]. The caspase-9 cleaves and activates down-regulated caspase-3. Caspase-3 serves as an essential mediator of apoptosis in the execution of the mitochondria-mediated apoptotic pathways [32].

In this study, our current results have proved that NMCP-2 can decrease the level of ROS and recover the decreased MMP caused by H_2_O_2_. Furthermore, images of fluorescence microscopy and TEM showed H_2_O_2_ treated groups exhibited the apoptosis morphology, such as chromatin condensation and ruptured cristae in the mitochondria, whereas NMCP-2 protected HEK 293T cells from the apoptosis phenotype induced by H_2_O_2_. Thus, we further investigated the effect of NMCP-2 on the mRNA expressions of apoptosis regulator genes (Bcl-2, Bax, caspase-9, and caspase-3) in the mitochondrial pathway using qPCR.

Figure 10 shows that caspase-9, Bax, caspase-3, and Bcl-2 in the mitochondrial apoptosis pathway were activated. Moreover, the mRNAs were expressed differently in the mitochondrial apoptosis pathway. The mRNA expressions of caspase-9, Bax, and caspase-3 were significantly higher in H_2_O_2_-treated cells compared with the control group. However, co-treatment with NMCP-2 (5, 25 µg/mL) suppressed the mRNA expression. The cells pretreated with NMCP-2 showed increased expression levels of Bcl-2 mRNA compared with the control group. Moreover, Bcl-2 showed higher mRNA expression in the H_2_O_2_-treated group. These results suggested that NMCP-2 reduced apoptosis through the mitochondrial pathway.

### 2.8. Effect of NMCP-2 on Cell Signals Related to Apoptosis Determined by Western Blot 

Previous studies suggested that the extracts from *M. conica* possessed antioxidant properties [12]. Crude polysaccharides from other fungus belonging to the genus *Morchella* prevented oxidative stress induced by H_2_O_2_ in cells [4]. Nevertheless, information on the molecular mechanism of purified polysaccharide from *M. conica* to protect the oxidative stress injury in vitro has been scarce. Thus, to further investigate the potential molecular mechanism, the expressions of proteins associated with the mitochondrial apoptotic pathway were detected by Western blot analysis.

As shown in Figure 11, after the treatment of HEK 293T cells with H_2_O_2_, the activities of Bax and Cleaved Caspase-3 were increased, whereas the expression of Bcl-2 was down-regulated. While the cells pretreated with NMCP-2 exerted a significant inhibitory effect on the levels of pro-apoptotic proteins Bax and Cleaved Caspase-3 expression, and the expression of anti-apoptotic protein Bcl-2 was up-regulated. Taken together, these findings show that the effect of NMCP-2 against H_2_O_2_-induced apoptosis was mediated by the regulation of Bax, Bcl-2, and Caspase-3 expression in the mitochondrial apoptosis pathway. 

## 3. Materials and Methods

### 3.1. Materials and Reagents

The fruiting bodies of *M. conica* were obtained from Changchun in Jilin Province (China). DEAE-52 cellulose, Sephadex G100, DPPH (1, 1-diphenyl-2-picrylhydrazyl), standard monosaccharides, 3-(4, 5-Dimethylthiazol-2-yl)-2, 5-diphenyltetra-zolium bromide (MTT), and dimethyl sulfoxide (DMSO) were purchased from Sigma–Aldrich Co., Ltd. (St. Louis, MO, USA). The Reactive Oxygen Species Assay kits were produced by Beyotime Inc. (Beyotime, Shanghai, China). TRIzol-A+ reagent, FastQuant RT Kits (With gDNase), and Quant qRT-PCR kits were obtained from Tiangen Inc. (Tiangen, Beijing, China). All the antibodies for Western blot analysis were provided by Cell Signaling Technology Inc. (Beverly, MA, USA). Other chemicals and reagents were of analytical grade.

### 3.2. Extraction and Purification of MCP

The extraction of crude MCP was carried out according to the method described by Xu et al. [33]. Briefly, the dried *M. conica* powder was defatted and decoloured in a Soxhlet apparatus, dried, and mixed with distilled water for extraction at the optimal extraction conditions (microwave power of 210.61 W, W/M ratio of 41.07:1, and extraction time of 126.98 s) in a UMSE(ultrasonic-microwave synergistic extraction) apparatus (XO-SM50, Nanjing Xianou Instrument Co. Ltd., Nanjing, Jiangsu, China). The obtained solution was concentrated with a rotary evaporator, precipitated with four volumes of 95% (*v*/*v*) ethanol, and centrifuged to collect the precipitate as crude polysaccharide. The crude polysaccharides were re-dissolved and centrifuged. The supernatant was applied to a DEAE-52 cellulose column (2.6 × 40 cm), equilibrated in distilled water, and eluted stepwise with a linear gradient of 0–0.25 M NaCl at the flow rate of 1 mL/min. The total polysaccharide content in each fraction was quantified by the phenol-sulfuric acid assay. One major MCP fraction, namely NMCP, was collected, concentrated, lyophilized, and further purified with Sephadex G100 column (2.6 × 50 cm) and eluted with distilled water at the flow rate of 0.5 mL/min. The fractions NMCP-1 and NMCP-2 were collected and lyophilized for further study.

### 3.3. Radical Scavenging Activity and Metal Chelating In Vitro

#### 3.3.1. DPPH Radical Scavenging Assay

The DPPH radical scavenging activity of polysaccharides was carried out by the method of [34] with some modifications. Briefly, 1 mL of each sample (0.1, 0.5, 1, 2, 3, and 4 mg/mL) was mixed thoroughly with 1 mL of ethanol solution containing DPPH (0.1 mM). After a 30-min incubation period at room temperature, the absorbance of the solution was measured at 517 nm. Vc was used as the positive control. Inhibition of free radicals by DPPH was calculated using to the following equation:DPPH radical scavenging activity (%) = (*A*_Control_ − *A*_Sample_/*A*_Control_) × 100 (1)
where *A*_Control_ is the absorbance of the control and *A*_sample_ is the absorbance of the samples. All tests were carried out at least three times.

#### 3.3.2. Ferrous Chelating Activity

Chelating ability was determined according to the method of [35]. One milliliter of the sample was mixed with 3.7 mL of deionized water. Then, the mixture reacted with 0.1 mL of FeCl_2_ (2 mM) and 0.2 mL of ferrozine (5 mM) for 20 min at room temperature. The absorbance was determined at 562 nm using EDTA as a reference. The chelating ability was calculated using the formula given below:Chelating activity (%) = (*A*_Control_ − *A*_Sample_/*A*_Control_) × 100(2)
where *A*_Control_ is the absorbance of EDTA as the positive control and *A*_Sample_ is the absorbance of the measured samples.

### 3.4. Preliminary Characterization of NMCP-2

#### 3.4.1. Fourier-Transform infrared (FT-IR) Spectroscopy Assay

The FT-IR spectrum was recorded by using an IR Prestige-21 Fourier-transform infrared (FT-IR) spectrometer (Shimadzu, Japan) in the range of 4000–400 cm^−1^. Briefly, 2 mg of the dried NMCP-2 was ground with 200 mg of KBr powder and pressed into 1-mm thick pellets for the analysis [36].

#### 3.4.2. Measurement of Molecular Weights

The molecular weight of NMCP-2 was measured by high performance liquid chromatography (HPLC) on an Agilent 1200 Series HPLC System equipped with an TSKgel^®^ G5000PWXL column (7.8 mm × 300 mm) and RI (refractive index) and UV detectors. The column was maintained at 20 °C and eluted with 0.002 mol/L sodium dihydrogen phosphate at a flow rate of 0.6 mL/min. The column was calibrated with a series of standard dextrans (1000, 5000, 25,000, 410,000, and 1,100,000 Da). The molecular weight of NMCP-2 was calculated by the calibration curve.

#### 3.4.3. Monosaccharide Composition Analysis

Determination and quantification of the monosaccharide composition was determined by gas chromatography-mass spectrometry (GC-MS). Briefly, 5 mg of NMCP-2 was hydrolyzed with 2 M trifluoroacetic acid (TFA) in a sealed tube at 99 °C for 5 h. The excess acid was removed with MeOH. The hydrolyzed products were reduced through the addition of NaBH_4_ at room temperature, acetylated with acetic acid, concentrated, and lyophilized [37]. Acetylation was conducted with pyridine and acetic anhydride.

The obtained alditol acetates were subjected to GC-MS using an Agilent 7890A-5975C instrument equipped with a DB-5 capillary column (30 m × 0.25 mm × 0.25 μm) and a flame ionization detector. The GC-MS was operated under the following chromatographic conditions: the injection and detection temperatures were set at 250 and 280 °C, respectively. The carrier gas was high purity nitrogen with a flow rate of 0.6 mL/min. The column temperature was held at 200 °C for 2 min, increased to 245 °C at the rate of 3 °C/min, then programmed to increase at 10 °C/min to 270 °C and then was held for 2 min.

#### 3.4.4. Nuclear Magnetic Resonance (NMR) Spectroscopy Analysis

NMCP-2 was exchanged with deuterium by freeze-drying against D_2_O three times. The ^1^H, ^13^C, HSQC, and HMBC NMR spectra were given on a Bruker Advance 600 MHz NMR spectrometer. ^1^H and ^13^C NMR spectra were conducted at 600 and 151.01 MHz, respectively [38]. HSQC and HMBC were carried out by the standard program.

### 3.5. Culture of HEK 293T Cell Lines

Human embryonic kidney (HEK) 293T cells (ATCC, Manassas, VA) were cultured in Dulbecco’s modified Eagle’s medium (DMEM) containing 10% fetal bovine serum (Hyclone), 100 U/mL penicillin, and 100 mg/mL streptomycin at 37 °C in a humid atmosphere containing 5% CO_2_.

### 3.6. Cell Viability Assay

Cell viability was determined using an MTT assay. HEK 293T cells were seeded at a density of 5 × 10^4^ cells/mL in 96-well cell culture plates and incubated for 24 h before two different experimental treatments were administered. The first group of cells was treated with a medium containing various concentration of NMCP-2 (0, 5, 10, 25, 50 μg/mL) for 24 h to determine whether NMCP-2 was toxic to the cells. The second group was pretreated with the same concentration of NMCP-2 as the first group and exposed to 400 μM H_2_O_2_ for 6 h. Afterwards, a total of 10 μL of MTT solution (5 mg/mL) was added. After further incubation for 4 h, 100 μL Formanzan dissolving solution was added and incubated for 4 h in order to solubilize purple formazan crystals inside the intact mitochondria. Absorbance at 570 nm was measured using a microplate reader (BioTek Instruments, Winooski, Vermont, USA) [39].

### 3.7. Measurement of Intracellular ROS Production

After treatment with the H_2_O_2_, the cells were incubated with 10 μM fluorescence probe DCFH-DA for 20 min at 37 °C. Then, the cells were washed thrice with phosphate-buffered saline (PBS). The fluorescence intensity was tested in a multi-mode microplate reader (BioTek Instruments, USA) with an excitation wavelength of 488 nm and an emission wavelength of 525 nm [40]. 

### 3.8. Determination of Mitochondrial Membrane Potential (MMP)

MMP of the cells was determined by measuring cell retention of the fluorescent cationic JC-1 (Beyotime, Shanghai, China). Cells with higher MMP contain aggregates of JC-1 emitting red fluorescence. When the ΔΨm dissipates, monomers of JC-1 are produced with green fluorescence. Cells were incubated with a JC-1 working solution at 37 °C for 20 min in the dark. Subsequently, the cells were washed twice with JC-1 staining buffer. The fluorescence was immediately recorded using a multi-mode microplate reader (BioTek Instruments, USA). The aggregates were detected with an excitation wavelength of 525 nm and an emission wavelength of 590 nm. The monomers were detected with an excitation wavelength of 490 nm and an emission wavelength of 530 nm [41]. 

### 3.9. Fluorescent Microscopy Measurements 

After the H_2_O_2_ challenge, the cells were stained with Hoechst 33342 at 4 °C for 20 min and rinsed with phosphate-buffered saline (PBS) to remove the dye. Cell morphology was taken by fluorescence microscopy (FV-1000 spectral, Olympus, Tokyo, Japan).

### 3.10. Transmission Electron Microscopy (TEM) Analyses

The pretreated cells were post-fixed in 2% osmium tetroxide. Following the dehydration in acetone, the samples were infiltrated and embedded in Epon 812 (Fluka Chemie AG, Buchs, St. Gallen Switzerland) by standard techniques. The thin sections (1 μm) were double stained with 1% toluidine. For electron microscope examination, the ultrathin sections (50 nm) were cut and stained with uranyl acetate and lead citrate. The stained ultramicrotomies were photographed by using a Philips CM-10 transmission electron microscope [42].

### 3.11. Quantitative Real-Time Polymerase Chain Reaction (q-PCR)

Total RNA isolation was performed using TRNzol reagent (TIANGEN, Beijing, China). The mRNA was treated with DNase I (Fermentas, Hanover, NH, USA) to remove the possible DNA residues. The mRNA was reverse transcripted into cDNA using the BioRT cDNA First-Strand Synthesis Kit (Bioer Technology, Hangzhou, China). q-PCR was conducted in a BIO-RAD Iq5 Multicolor Real-Time PCR Detection System using the BioEasy SYBR Green I Real Time PCR Kit (Bioer Technology, Hangzhou, China). The q-PCR assay was performed three times [43]. The primer sequences are listed in Table 2.

### 3.12. Western Blotting Analysis

The cells were harvested in lysis buffer and the protein concentrations were determined by the Bradford method. The protein samples were separated by SDS-polyacrylamide gel electrophoresis and transferred to PVDF (polyvinylidene difluoride) membranes using the semi-dry transfer method. The membranes were incubated with the primary antibodies (Bax, Bcl-2, caspase-3, and β-actin) at 1:1000 dilutions. Immune complexes were incubated with horseradish peroxidase (HRP) conjugated secondary antibodies at 1:2000 dilution at room temperature for 1 h. The blots were incubated, exposed, and analyzed. The band intensities were quantified using ImageJ software [44].

### 3.13. Statistical Analysis

All data were analyzed using SPSS 20.0 (SPSS, Chicago, IL, USA) software. All values are expressed as means ± standard deviation (SD). A one-way ANOVA performed by Dunnett’s test was used for statistical comparison among the multiple groups, where *p* < 0.05 was considered as significant. Statistical significance between two groups was analyzed using the Student *t*-test. A value of *p* < 0.05 (* *p* < 0.05, ** *p* < 0.01) between test groups was considered as statistically significant.

## 4. Conclusions

In the present study, we reported antioxidant activity of the polysaccharide NMCP-2 isolated from *M. conica*. NMCP-2 prevented oxidative stress induced by H_2_O_2_ in human embryonic kidney (HEK) 293T cells, ameliorated mitochondrial function, and reduced cell apoptosis. NMCP-2 suppressed overexpression of apoptosis-promoting signals (Bax, caspase 3) in H_2_O_2_-induced HEK 293T cells, while preventing downregulation of anti-apoptotic protein (Bcl-2). These data suggest that NMCP-2 prevents apoptotic cell death of HEK 293T cells via the mitochondria-mediated cell apoptosis pathway. This study indicates that NMCP-2 can be further explored as a potential natural antioxidant for oxidative diseases.

## Figures and Tables

**Figure 1 ijms-19-04027-f001:**
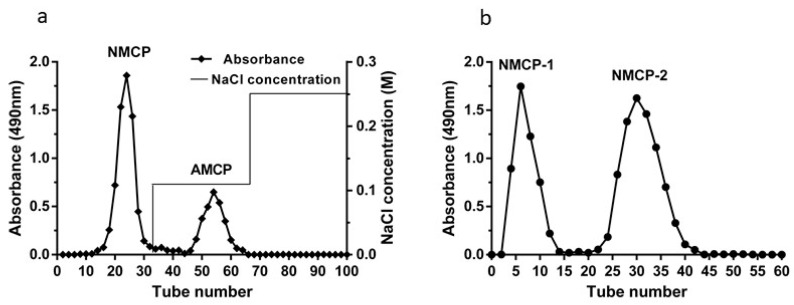
The elution curve of polysaccharides isolated from the *M. conica* on a DEAE-52 cellulose column. (**a**) The DEAE-52 cellulose column was eluted with a 0–0.25 mol/L linear gradient of NaCl at a flow rate of 1 mL/min. The polysaccharide fractions were pooled and named as neutral *M. conica* polysaccharides (NMCP) and AMCP, respectively. (**b**) Elution curve of the NMCP on a Sephadex G-100 column. The Sephadex G-100 column was eluted with distilled water at a flow rate of 0.3 mL/min. The two polysaccharide fractions were named NMCP-1 and NMCP-2, respectively.

**Figure 2 ijms-19-04027-f002:**
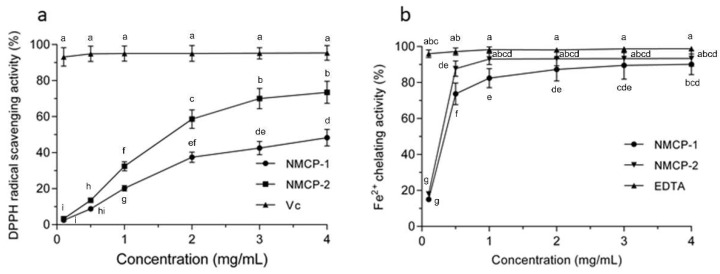
Antioxidant activity of NMCP-1 and NMCP-2 in vitro. (**a**) 2,2-diphenyl-1-picrylhydrazyl (DPPH) radical-scavenging activity. Vc was used as a positive control. (**b**) Chelating activity on Fe^2+^. EDTA (Ethylenediaminetetraacetic acid) was used as reference standard. Results are presented as means ± standard deviations (*n* = 3). Different superscripts (a–i) within the same figure are significantly different (*p* < 0.05).

**Figure 3 ijms-19-04027-f003:**
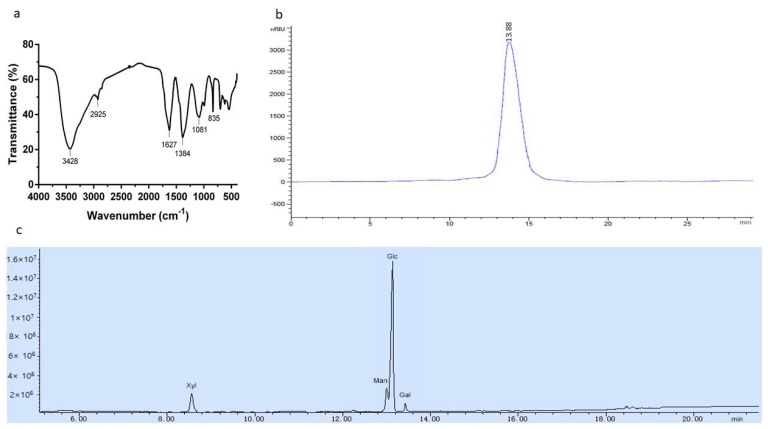
(**a**) FT-IR spectra of NMCP-2. NMCP-2 was measured in the range of 4000–400 cm^−1^. (**b**) Molecular weights distribution of NMCP-2. The column was maintained at 20 °C and eluted with 0.002 mol/L sodium dihydrogen phosphate at a flow rate of 0.6 mL/min. (**c**) Monosaccharide composition of NMCP-2 using gas chromatography-mass spectrometry (GC-MS).

**Figure 4 ijms-19-04027-f004:**
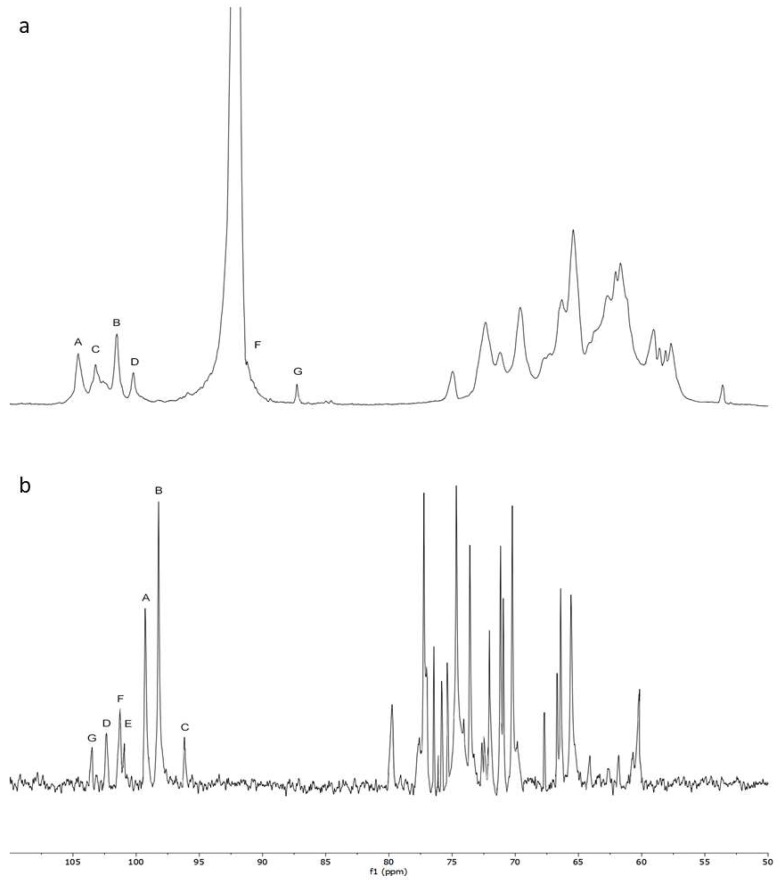
(**a**) ^1^H NMR spectrum, (**b**) ^13^C NMR spectrum, (**c**) HSQC NMR spectrum, and (**d**) HMBC NMR spectrum of polysaccharide fraction NMCP-2 recorded in D_2_O.

**Figure 5 ijms-19-04027-f005:**
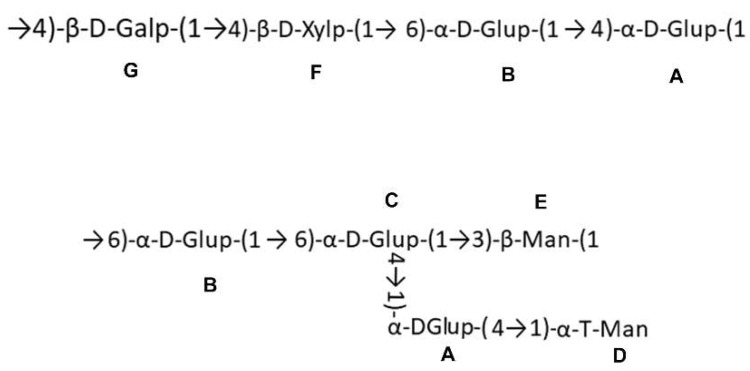
Putative fragments of the structure of NMCP-2. A–G are units of glycosyl residues.

**Figure 6 ijms-19-04027-f006:**
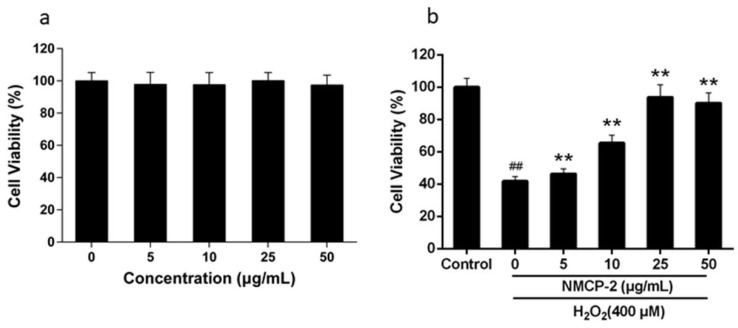
Effects of NMCP-2 and H_2_O_2_ on the cell viability of HEK 293T cells. (**a**) Effects of NMCP-2 on the HEK 293T cell survival rate. (**b**) Protective effects of NMCP-2 against oxidative damage induced by H_2_O_2_. Data are expressed as the mean ± standard deviation. ^##^
*p* < 0.01 differs from control group; ** *p* < 0.01 differs from model group.

**Figure 7 ijms-19-04027-f007:**
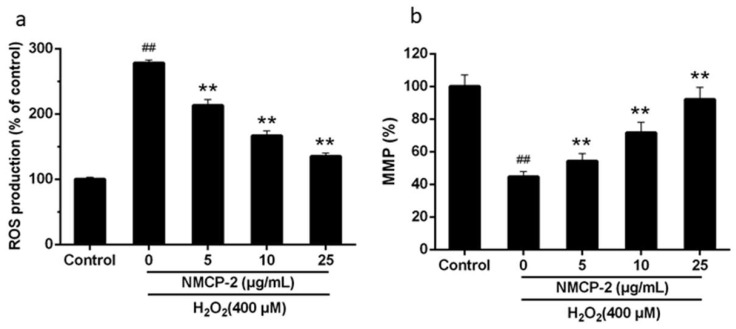
Effect of NMCP-2 on H_2_O_2_-induced intracellular accumulation of reactive oxygen species (ROS) and mitochondria membrane potential (MMP). Histogram showing the (**a**) ROS level and (**b**) MMP level in HEK 293T cells after treatment with H_2_O_2_ in presence or absence of NMCP-2 (5, 25 μg/mL) compared to untreated groups. Data are presented as mean ± SD (*n* = 3). ^##^
*p* < 0.01 differs from control group; ** *p* < 0.01 differs from model group.

**Figure 8 ijms-19-04027-f008:**
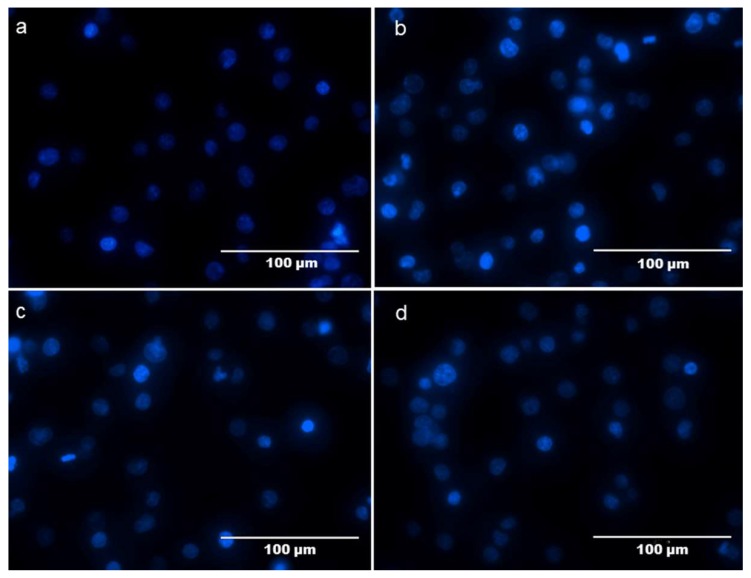
Fluorescent photo micrographs after staining with Hoechst 33342. (**a**) Untreated control. (**b**) Cells exposed to 400 μM of H_2_O_2_. (**c**,**d**) Cells were treated with NMCP-2 (5, 25 μg/mL) for 24 h followed by exposure to 400 μM H_2_O_2_.

**Figure 9 ijms-19-04027-f009:**
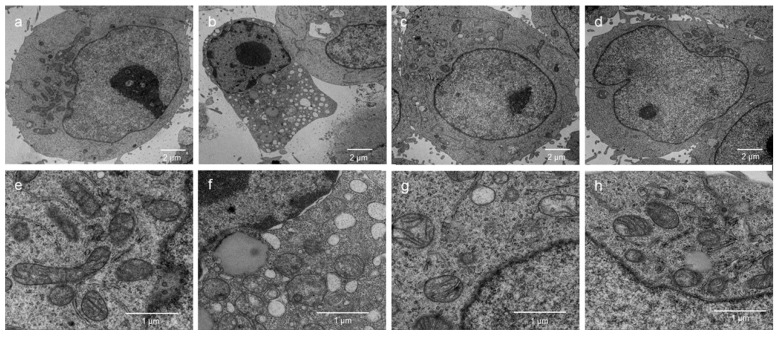
The effect of NMCP-2 on the ultrastructural changes of HEK 293T cells exposed to H_2_O_2_. Transmission electron micrograms show the ultrastructure of HEK 293T cells before (**a**,**e**) and after (**b**,**f**) H_2_O_2_ injury or treatment with 5 μg/mL NMCP-2 (**c**,**g**) or 25 μg/mL NMCP-2 (**d**,**h**) after the injury.

**Figure 10 ijms-19-04027-f010:**
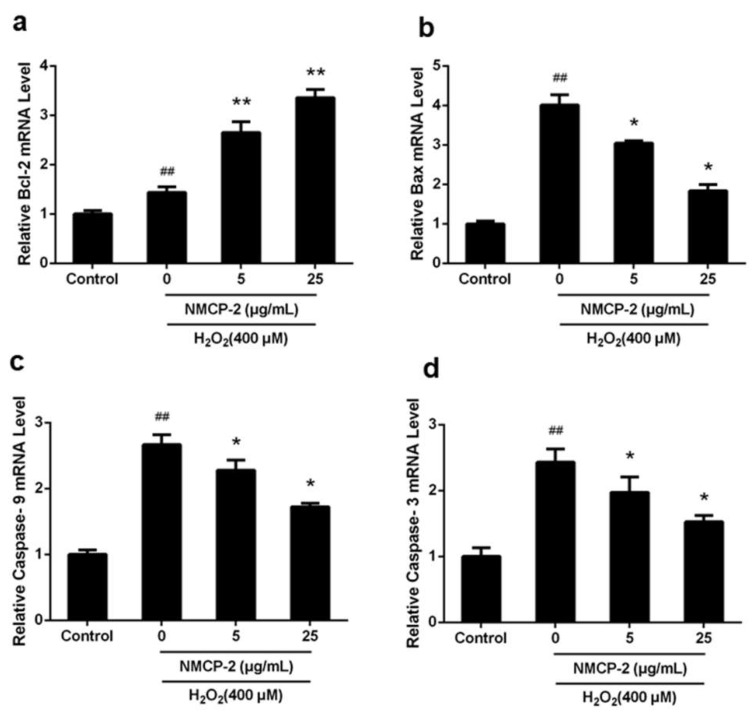
The mRNA expression of genes in the mitochondrial apoptosis pathway. Real-time qPCR analysis of (**a**) Bcl-2, (**b**) Bax, (**c**) Caspase-9, and (**d**) Caspase-3. Data are presented as mean ± SD (*n* = 3). ^##^
*p* < 0.01 differs from control group; * *p* < 0.05 differs from model group; ** *p* < 0.01 differs from model group.

**Figure 11 ijms-19-04027-f011:**
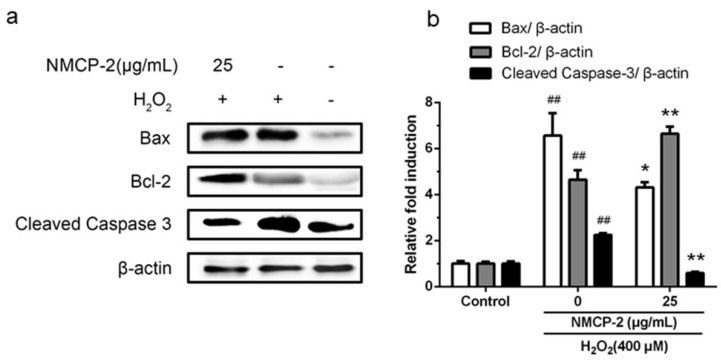
(**a**) Effect of NMCP-2 on Bax, Bcl-2, and caspase-3 activities of HEK 293T cells. (**b**) Bax, Bcl-2, and caspase-3 expression were quantified by densitometry and normalized to β-actin levels. Results are presented as mean ± SD of three separate experiments. ^##^
*p* < 0.01 differs from control group; * *p* < 0.05 differs from model group; ** *p* < 0.01 differs from model group.

**Table 1 ijms-19-04027-t001:** The ^1^H and^13^C NMR shifts of NMCP-2 in D_2_O at 298 K.

Residues	Sugar linkage		1	2	3	4	5	6
**A**	→4)-α-D-Glup-(1→	H	5.12	3.59	3.93	3.81	3.56	3.96
		C	99.30	71.16	71.90	76.48	75.31	61.02
**B**	→6)-α-D-Glup-(1→	H	5.02	3.60	3.73	3.56	3.96	4.05
		C	98.22	77.26	74.15	70.35	71.16	66.34
**C**	→4, 6)-α-D-Glup-(1→	H	5.08	-	3.96	3.69	-	3.69
		C	96.24	72.54	74.15	77.73	-	63.33
**D**	α-T-Manp	H	4.98	4.00	3.72	3.57	3.69	3.69
		C	102.22	70.05	69.87	66.86	73.29	60.86
**E**	→3)-β-Manp-(1→	H	-	4.13	3.96	3.69	3.42	3.81
		C	101.15	71.16	74.31	67.82	76.08	61.55
**F**	→4)-β-D-Xylp-(1→	H	4.67	3.56	3.69	3.85	4.13	-
		C	101.30	79.61	75.69	77.01	66.86	-
**G**	→4)-β-D-Galp-(1→	H	4.54	3.33	3.56	3.72	-	-
		C	103.43	73.29	76.67	76.38	-	-

“-” means not obtained because of low resolution.

**Table 2 ijms-19-04027-t002:** Primers used for qPCR.

Name	Primers	Sequence (5’-3’)
**CASPASE-3**	CASPASE-3F	AACTGGACTGTGGCATTGAG
	CASPASE-3R	ACAAAGCGACTGGATGAACC
**CASPASE-9**	CASPASE-9F	GGAAGAGGGACAGATGAATG
	CASPASE-9R	TTGTTTGGCACCACTCAG
**BAX**	BAX-F	CCTTTTCTACTTTGCCAGCAAAC
	BAX-R	GAGGCCGTCCCAACCAC
**BCL-2**	BCL-2F	CGGTTCAGGTACTCAGTCATC
	BCL-2R	CGGTGGGGTCATGTGTGTG
**GAPDH**	GAPDH-F	GTCAACGGATTTGGTCGTA
	GAPDH-R	GTAGTTGAGGTCAATGAAG

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
