# Peer review of "A Polysaccharide Purified from Morchella conica Pers. Prevents Oxidative Stress Induced by H2O2 in Human Embryonic Kidney (HEK) 293T Cells"

_ijms, 2018, doi:10.3390/ijms19124027_

Reviewer 1 Report

This excellent work explores the effect of the fungus polysaccharide NMCP-2 on HEK-293T cells. NMCP-2 reduced oxydative stress induced by H2O2 treatment. NMCP-2 reduced apoptosis and overexpression of Bac and caspase 3. NMCP-2 protected HEK-293T after H2O2 treatment from mitochondria-mediated apoptosis. These data indicate that NMCP-2 should be explored as a potential anti-oxidant.

All results are credible and conclusion drawn are convincing. The linguistic style is very good making the manuscript easy to read and understand. This manuscript will find the interest of many readers. Therefore I strongly support publication.

I just found one mistake that should be corrected:

L190: Change "(MCPN)" to "(NMCP)"

Author Response

Point 1: I just found one mistake that should be corrected: L190: Change "(MCPN)" to "(NMCP)". 

Thank you for your suggestion. We have Change "(MCPN)" to "(NMCP)"

Line 61

The main fraction (NMCP)

Reviewer 2 Report

Review of Manuscript ID: IJMS-403354

Title: A polysaccharide purified from Morchella conica prevents oxidative stress induced by H2O2 in human embryonic kidney (HEK) 293T cells

In recent years, some bioactive polysaccharides isolated from natural sources have attracted much attention in the field of biochemistry and pharmacology. In particular, polysaccharides isolated from fungi, including those from Morchella conica, have been shown to be effective antioxidants. Therefore, the direction of research undertaken by the authors of the manuscript is very interesting both for application and cognitive reasons.

I fully recommend the manuscript to be published after taking into account minor comments.

Abstract

L2 “……Morchella conica” replace “…Morchella conica”

L3 “….H2O2” replace “….H2O2”

L30  fill up your keywords

L53  please explain shortcuts –MCP and NMCP-2

L66 expand the description of the methodology

L94, 119, 128, 139, 144, 157, 164, 172 - no literature reference

L201 - compare the antioxidant capacity of NMCP-2 polysaccharides with other polysaccharides isolated from fungi

L302 - “……photom icographs” replace “…photo micrographs”

In verses 73, 74, the authors of the manuscript write that they received two MCP polysaccharide fractions. Why in that case, they only analyzed NMCP-2, and did not deal with NMCP-1? What was the reason?

Author Response

The line numbers are renumbered after revision. The respond is according to new line numbers.

Point 1: L 2 “……Morchella conica” replace “…Morchella conica”. 

Response 1: Thank you for pointing it out. We’ve change them into Italic.

Line 2 Morchella conica

Point 2: L3 “….H2O2” replace “….H2O2”.

Response 2: Thank you. We’ve changed H2O2 into correct form.

Line 3 H2O2

Point 3: L30  fill up your keywords

Thank you so much. We’ve fill up the keywords.

Line 31 Keywords: Morchella conica; polysaccharides; oxidative stress

Point 4: L53  please explain shortcuts –MCP and NMCP-2

Thank you for your suggestions. We’ve explain the shortcuts of MCP and NMCP-2 when they first show up in this article.

Line 16 NMCP-2 (neutral M. conica polysaccharides-2)

Line 55 MCP (M. conica polysaccharides)

Point 5: L66 expand the description of the methodology

Thank you for pointing it out. We’ve expand the description of the methodology.

Line 284 The extraction of crude MCP was carried out according to the method described by Xu et al. [33]. Briefly, the dried M. conica powder was performed in a Soxhlet apparatus, dried and mixed with distilled water for extraction at the optimal extraction conditions (microwave power of 210.61 W, W/M ratio of 41.07:1, and extraction time of 126.98 s) in an UMSE apparatus (XO-SM50, Nanjing Xianou Instrument Co. Ltd., China). The obtained solution was concentrated with a rotary evaporator, precipitated with four volumes of 95% (v/v) ethanol and centrifuged to collect the precipitate as crude polysaccharide. The crude polysaccharides were re-dissolved and centrifuged. The supernatant was applied to a DEAE-52 cellulose column (2.6Ă—40 cm), equilibrated in distilled water and eluted stepwise with a linear gradient of 0-0.25 M NaCl at the flow rate of 1 mL/min. The total polysaccharide content in each fraction was quantified by the phenol-sulfuric acid assay. One major MCP fraction namely NMCP were collected, concentrated, lyophilized, and further purified with Sephadex G100 column (2.6Ă—50 cm), eluted with distilled water at the flow rate of 0.5 mL/min. The fractions NMCP-1 and NMCP-2 were collected and lyophilized for further study.

Point 6: L94, 119, 128, 139, 144, 157, 164, 172 - no literature reference

Thank you for your suggestion. We’ve added the references separately.

Line 316

3.4.1. Fourier-transform infrared (FT-IR) spectroscopy assay

The FT-IR spectrum was recorded by using a IR Prestige-21 Fourier-transform infrared (FT-IR) spectrometer (Shimadzu, Japan) in the range of 4000-400 cm-1. Briefly, 2 mg of the dried NMCP-2 was ground with 200 mg of KBr powder and pressed into 1 mm thick pellet for the analysis [36]. 

Line 341

3.4.4. Nuclear magnetic resonance (NMR) spectroscopy analysis

NMCP-2 was exchanged with deuterium by freeze-drying against D2O for three times. The 1H, 13C, HSQC, HMBC NMR spectra were given on a Bruker Advance 600 MHz NMR spectrometer. 1H, 13C NMR spectra were conducted at 600 MHz and151.01 MHz, respectively [38]. HSQC and HMBC were carried out by the standard program

Line 350

3.6. Cell viability assay

Cell viability was determined using MTT assay. HEK 293T cells were seeded at a density of 5 Ă— 104 cells/mL in 96-well cell culture plates and incubated for 24 h before two different experimental treatments were administered. The first group of the cells was treated with a medium containing various concentration of NMCP-2 (0, 5, 10, 25, 50 ÎĽg/mL) for 24 h to determine whether NMCP-2 was toxic to the cells. The second group was pretreated with the same concentration of NMCP-2 as the first group and exposed to 400 ÎĽM H2O2 for 6 h. Afterwards, a total of 10 ÎĽL of MTT solution (5 mg/mL) was added. After further incubation for 4 h, 100 ÎĽL Formanzan dissolving solution was added and incubated for 4 h in order to solubilize purple formazan crystals formed inside the intact mitochondria. Absorbance at 570 nm was measured using a microplate reader (BioTek Instruments,USA) [39].

Line 361

3.7. Measurement of intracellular ROS production

After treatment with the H2O2, the cells were incubated with 10 μM fluorescence probe DCFH-DA for 20 min at 37 °C. Then the cells were washed thrice with PBS. The fluorescence intensity was tested in a multi-mode microplate reader (BioTek Instruments, USA) with an excitation wavelength of 488nm and an emission wavelength of 525nm [40].

Line 366

3.8. Determination of mitochondrial membrane potential (MMP)

MMP of the cells was determined by measuring cell retention of the fluorescent cationic JC-1 (Beyotime, Shanghai, China). Cells with higher MMP contain aggregates of JC-1 emitting red fluorescence. When the ΔΨm dissipates, monomers of JC-1 were produced with green fluorescence. Cells were incubated with JC1 working solution at 37 °C for 20 min in the dark. Subsequently, the cells were washed twice with JC-1 staining buffer. The fluorescence was immediately recorded using a multi-mode microplate reader (BioTek Instruments, USA).  The aggregates were detected with an excitation wavelength of 525 nm and an emission wavelength of 590 nm. The monomers were detected with an excitation wavelength of 490 nm and an emission wavelength of 530 nm [41].

Line 379

3.10. Transmission electron microscopy (TEM) analyses 

The pretreated cells were postfixed in 2% osmium tetroxide. Following the dehydration in acetone, the samples were infiltrated, and embedded in Epon 812 (Fluka Chemie AG) by standard techniques. The thin sections (1 ÎĽm) were double stained with 1% toluidine. For electron microscope examination, the ultrathin sections (50 nm) were cut and stained with uranyl acetate and lead citrate. The stained ultramicrotomies were photographed by using Philips CM-10 transmission electron microscope [42].

Line 386

3.11. Quantitative real-time polymerase chain reaction (q-PCR)

Total RNA isolation was performed using TRNzol reagent (TIANGEN, Beijing, China). The mRNA was treated with DNase I (Fermentas) to remove the possible DNA residues. The mRNA was reverse transcripted into cDNA using the BioRT cDNA First-Strand Synthesis Kit (Bioer Technology, Hangzhou, China). q-PCR was conducted in a BIO-RAD Iq5 Multicolor Real-Time PCR Detection System using the BioEasy SYBR Green I Real Time PCR Kit (Bioer Technology, Hangzhou, China).  The q-PCR assay was performed with 3 repeats [43]. The primer sequences are listed in Table 1.

Line 394

3.12. Western blotting analysis

The cells were harvested in lysis buffer and the protein concentrations were determined by Bradford method (Bio-Rad). The protein samples were separated by a SDS-polyacrylamide gel electrophoresis and transferred to PVDF membranes using the semi-dry transfer method. And the membranes were incubated with the primary antibodies (Bax, Bcl-2, caspase-3 and β-actin) at 1:1000 dilutions. Immune complexes were incubated with horseradish peroxidase (HRP) conjugated secondary antibodies at 1:2000 dilution at room temperature for 1 h. The blots were incubated, exposed and analyzed. The band intensities were quantified using ImageJ software [44].

Point 7: L201 - compare the antioxidant capacity of NMCP-2 polysaccharides with other polysaccharides isolated from fungi

Thank you for pointing this out. We’ve added the comparison in the corresponding paragraphs.

Line 72

2.2. DPPH scavenging effect and ferrous ion chelating ability of NMCP-1 and NMCP-2

DPPH is one of a stable free radical that has been extensively used for free radical elimination reaction. Because the free radical would be scavenged when it encounters an electron or hydrogen donor [13]. It could be seen from Figure 2a that the DPPH radical scavenging abilities of NMCP-1, NMCP-2 showed a dose-dependent manner by comparison with the same concentrations of Vc. At the concentration of 4 mg/mL, the scavenging activities of NMCP-1, NMCP-2 are 48.29±4.61 %, 73.49± 6.14 %, respectively. The DPPH scavenging ability in NMCP-2 at six concentrations from 0.1 to 4 mg/mL was significantly stronger than that in NMCP-1 groups at the same concentration (P<0.05). Compared with other polysaccharides purified from fungi, the DPPH scavenging ability of NMCP-2 is similar to GFP-2 purified from Grifola frondosa. At the concentration of 3 mg/mL, the DPPH scavenging ability of GFP-2 and NMCP-2 were 79.6% and 70%, respectively [14]. While the DPPH scavenging ability of NMCP-2 is higher than the polysaccharides purified from Penicillium sp. F23-2 [15]. In our study, it was found that NMCP-1 and NMCP-2 are hydrogen donors to the DPPH free radicals, thereby terminating the radical chain reaction. NMCP-2 showed remarkable scavenging capacities than NMCP-1 at the dosage of 0–4 mg/mL.

Ferrous is the strongest prooxidant on stimulating the lipid peroxidation among transition metals. Hence, the Fe2+ chelating capacity was applied to antioxidant research via a measurement of the iron-ferrozine complexes [16]. As shown in Fig 2b, the Fe2+ chelating rates of purified NMCP-1increased from 15.02 % to 90.15% when the concentration increased from 0.1 to 4.0 mg/mL. For NMCP-2, the chelating ferrous ability increased from 18.24% to 93.08% as the concentration increased from 0.1 to 1.0 mg/mL, and slightly increased when the concentration of NMCP-2 was from 2.0 to 4.0 mg/mL. The Fe2+ chelating capacity in NMCP-2 at 0.5 and 1 mg/mL groups was significantly stronger than that in NMCP-1 groups at the same concentration (P<0.05). Furthermore, the NMCP-2 possessed superior binding capacity for Fe2+ than NMCP-1. NMCP-2 was also more effective in chelating ability than other fungi polysaccharides, such as Tricholoma matsutake [17]. Because NMCP-2 possessed higher antioxidant activity of DPPH scavenging and ferrous ion chelating ability than NMCP-1, NMCP-2 was selected for the subsequent assay[18,19].

Point 8: L302 - “……photom icographs” replace “…photo micrographs”

Thank you for pointing this out. We’ve corrected the spell of the words.

Line 79

Figure 7. Fluorescent photo micrographs after staining with Hoechst 33342.

Point 9: In verses 73, 74, the authors of the manuscript write that they received two MCP polysaccharide fractions. Why in that case, they only analyzed NMCP-2, and did not deal with NMCP-1? What was the reason?

Thank you for your good question. We mentioned the reason of choosing NMCP-2 in the article, but not clearly enough. Thus, we corrected the sentences.

Line 96

Because NMCP-2 possessed higher antioxidant activity of DPPH scavenging and ferrous ion chelating ability than NMCP-1, NMCP-2 was selected for the subsequent assay [18,19].

Reviewer 3 Report

Overall, an excellent multiproxy research paper is submitted. There is no need of improvement of the English version of the text. All figures are informative and well presented. I just would like to suggest to include the author name of the fungal species in the beginning (non-obligatory): Morchella conica Pers. All other is fine.

Author Response

Point 1:I just would like to suggest to include the author name of the fungal species in the beginning (non-obligatory): Morchella conica Pers. All other is fine.. 

Response 1:

Thank you for pointing this out. We’ve added the author name in the beginning of the article.

Line 2

A polysaccharide purified from Morchella conica Pers. prevents oxidative stress induced by H2O2 in human embryonic kidney (HEK) 293T cells

Line 13

Abstract: Morchella conica Pers. (M. conica) has been used both as a medical and edible mushroom